# Dynamic Response of Atmospheric and Ocean Parameters and Their Relation to Typhoon Haikui (2012) Using Satellite Data

**Wangyuan Zhu** [1,2], **Mantravadi Venkata Subrahmanyam** [1,*], **Liuzhu Wang** [1] **and Biyun Guo** [1,3,*]

1 School of Marine Science and Technology, Zhejiang Ocean University, Zhoushan 316022, China
2 Zhejiang Haida Marine Survey, Planning and Design Co., Ltd., Zhoushan 316022, China
3 State Key Laboratory of Hydroscience and Engineering, Tsinghua University, Beijing 100084, China
* Correspondence: mvsm.au@gmail.com (M.V.S.); biyunguo@zjou.edu.cn (B.G.)

**Abstract:** Typhoon Haikui (2012) occurred in the northwestern Pacific Ocean, and landfall on the east coast of China brought heavy rainfall with strong winds. Because of Typhoon Haikui, sea surface temperature (SST) cooling of 3 °C occurred on the right side of the track, mainly due to Ekman transport and upwelling. SST cooling on the left side was lower than on the right side, mainly due to the rainfall. Heavy precipitation occurred on both sides of the typhoon track; however, rainfall was higher on the left side of the typhoon track. This paper explains the dynamic process between atmospheric and oceanographic parameters and verifies the variations in chlorophyll and sea surface height data before, during, and after the typhoon. Typhoon Haikui demonstrates dynamic variations and intuitively illustrates the relationship between the ocean and atmospheric parameters.

**Keywords:** typhoon; sea surface temperature (SST); wind; rainfall; chlorophyll; sea surface height (SSH)





## 1. Introduction

A typhoon is a mature tropical cyclone that forms on the tropical or subtropical sea at a surface temperature over 26 °C–27 °C [1]. Generally, in the summer season, the northwestern Pacific Ocean will generate a lot of violent storms called a typhoon. Some will disappear into the sea, and others landfall over the coast to produce a drastic impact through powerful winds and torrential rain [2]. Because of the severe damage caused by typhoons, meteorologists and oceanographers have been studying the cause and influence of typhoons extensively. Ocean–atmosphere interactions in cold sea surface temperature (SST) regions formed in the trail of two typhoon events [3]. Typhoons can induce cooling wakes at the ocean surface, causing lower SSTs along their track [4]. Different SSTs near the typhoon center result in differences in the atmospheric wind field and influence the intensity and typhoon track to some extent, with a significant impact on precipitation and latent heat flux near the eye [5–7]. The high-resolution WRF (Weather Research and Forecast) model coupled with the POM (Princeton Ocean Model) reproduces the features of the ocean [8]. Typhoons obtain energy from the warmer ocean and intensify if higher heat and moisture fluxes subsist [9]. Typhoons are illustrated by intense cyclonic winds, well-organized deep convection, and spiral rain bands.

A typhoon's induced rainfall magnitude and distribution are often multiscale and affected by many factors, such as storm size, track, translation speed, etc. Rodgers et al. [10] did quantitative research on the contribution of the North Atlantic Tropical Cyclone to rainfall. The objective method they used to separate tropical cyclone precipitation was with a fixed radius (444 km) around the center of a tropical cyclone. Yao Lina et al. [11] also used this method to analyze the precipitation after Typhoon Haikui faded. SST increases before a typhoon whereas the precipitation rate increases after a typhoon passes, which means the SST increase before typhoon occurrence prepares the required energy and causes additional precipitation in subsequent days [12]. Rainfall associated with a typhoon can become asymmetric after landfall. Typhoon landfall precipitation forecast is complex due

to the coastal and inland topography, land surface, and boundary layer conditions [13,14]. The precipitation that occurs due to typhoons will be variable. The rainfall on the left side of the typhoon track is higher than on the right side of the track [15], which enhances cooling on the left side. Owing to the heavy wind speed around the typhoon, higher wind stress will produce upwelling, and Ekman transport will be higher.

Surface wind stress curl produces upward motion of subsurface waters and surface waters to be transported away from the typhoon center, this is called Ekman transport. The wind stress is proportional to Ekman transport, and the sine of the latitude is inversely proportional [16]. Wind stress curl stands to have a significant impact on coastal upwelling [17]. The balance between the Coriolis force and friction is related to the Ekman dynamics. The sea level declines near the coast due to offshore Ekman transport [18]. During the cyclone passage, typhoon-induced upwelling by Ekman pumping and vertical mixing also acts [19]. Upwelling brings the subsurface nutrient waters to the surface, resulting in phytoplankton increase after a typhoon passes. Typhoon passage can also result in phytoplankton flourishing, which enhances chlorophyll concentration [20,21]. As we all know, ocean phytoplankton production plays a considerable role in the ecosystem, and primary production can be indexed by using chlorophyll-a (Chl-a) concentration [22]. Altimeter-measured sea surface height (SSH) variations collected by satellite present an excellent opportunity to observe the behavior of the ocean in the continental shelf region. The open ocean implications of the altimeter data are well understood [23], and this has been exploited in many studies to examine ocean circulation. Because wind stress forcing is a stochastic process, the average SSH response can be calculated with simultaneous wind forcing. The typhoon response is consistent with an Ekman transport moving to the right of the wind stress, generating an SSH gradient in the direction of the Ekman transport. The SSH response depends on wind stress in both space and time [24]. Satellite-derived SSH is used to quantify typhoon-induced ocean mixing and is closely related to SSH changes [25].

Typhoons are so transient and unpredictable that upper ocean responses are hard to capture using ship-borne observations and cruise tracks. Therefore, satellite observations and reanalysis data are used to investigate the dynamic processes of typhoon passage, as well as the location of an upper ocean response to a typhoon. Satellite remote sensing is an efficient and reliable way to provide continuous measurements of typhoon periods [2]. So, by understanding the relationship between typhoons and SST, chlorophyll concentration, wind, and precipitation, we can effectively forecast and reduce the damage caused over the coastal area. The typhoon's impact on SST can improve the productivity of sea areas. Thus, we can achieve the aim of effectively utilizing the marine environment. This paper deals with satellite data to analyze the dynamic processes during the time of Typhoon Haikui and their relation to each other. In addition, the post-typhoon effect on the enhancement of chlorophyll bloms was studied.

## 2. Materials and Methods

### 2.1. Typhoon Haikui Track Details

Typhoon Haikui (Number 11) was the strongest typhoon that landed in China in 2012. On 2 August, Haikui Typhoon originated at 22.4° N; 146.5° E and then intensified into a severe typhoon early on 7 August and, at 19:20 UTC (03:20 CST) on 8 August, typhoon Haiku made landfall over Xiangshan County in Zhejiang, China. The translation speed was not constant. Haikui moved with a faster translation speed, at 30 km/h, to the west. Later, the translation speed slowed down, and even had the three stagnations. Before landfall, translation speed accelerated, the typhoon developed into a severe tropical storm, and intensity increased with maximum wind speed near the center of 50.9 m/s. Typhoon Haikui weakened to a severe tropical storm and typhoon landfall was on 8 August on China's east coast. After landfall, Haikui moved westward along the stable path [26] through Ningbo, Shaoxing, and Hangzhou. After the typhoon's landfall the intensity increased—the first typhoon that had rapid onshore development since Typhoon Bill. The typhoon track of Haikui is given in the following Figure 1.

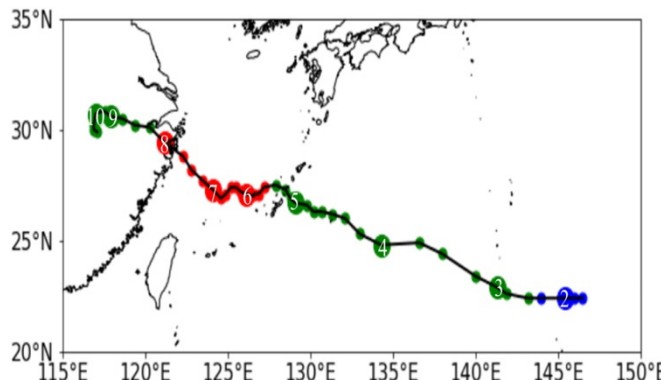

**Figure 1.** Typhoon Haikui track. The line represents the typhoon track, and the numbers in a circle are the dates of August 2012. The color circles are blue (lower intensity) to red (higher intensity).

Data

The study period chosen for Typhoon Haikui was between 2nd August and 11th August 2012. Haikui typhoon track position and intensities every 6 h were obtained from the JTWC (Joint Typhoon Warning Center) and typhoon information produced by the Japan Meteorological Agency (JMA) best-track dataset [27]. The following data sets were utilized to analyze dynamic variations due to Typhoon Haikui and to understand the relationship between atmospheric and ocean parameters:

(1) Daily SST data were from NOAA SST/OISST_AVHRR. OISST is the abbreviation of optimum interpolation sea surface temperature [8]. It is ideal for research activities where complete, daily SST data is more desirable than one with missing data due to orbital gaps or environmental conditions precluding SST retrieval. The data has a spatial grid resolution of 0.25°.

(2) Daily averaged wind data were from MERRA Daily Averaged v5.2.0. MERRA is a NASA reanalysis for the satellite era using a new version of the Goddard Earth Observing System Data Assimilation System Version 5 (GEOS-5). The project focuses on historical analyses of the hydrological cycle on a broad range of weather and climate time scales and places the NASA EOS suite of observations in a climate context [28].

(3) Wind stress data were from ASCAT, the Advanced SCATterometer. ASCAT scatterometer represents the latest implementation of space-borne microwave wind-measuring scatterometry. The Advanced SCATterometer (ASCAT) is on board the Metop-A satellite. The prime objective of ASCAT is to measure wind speed and direction over the ocean. The instrumental description of ASCAT can be obtained from [29,30]. ASCAT uses an objective method to estimate daily averaged wind and wind stress fields over global oceans with spatial resolutions of 0.25°.

(4) Daily precipitation data were collected from the merged GPCP (Global Precipitation Climatology Project). GPCP was established by the World Climate Research Program (WCRP) and Global Energy and Water Exchange (GEWEX) to develop a complete understanding of spatial and temporal patterns of global precipitation. GPCP provides community global precipitation products with satellite and daily rain gauge information [31].

(5) The most typical optical sensors are used in the Moderate Resolution Imaging Spectroradiometer (MODIS) sensor used for chlorophyll surveys with global algorithms [32]. Weekly chlorophyll data were from MODIS Aqua Chlorophyll-a level 3. The NASA Aqua satellite carries the MODIS sensor to observe chlorophyll-a concentrations in global oceans. The Level 3 standard mapped image (SMI) chlorophyll-a dataset has a weekly temporal resolution [32].

(6) Weekly SSH data were acquired from AVISO TOPEX/ERS/Jason1 merged. The "sea surface height" [33–35] is a sea surface height anomaly, in which "sea surface height" at a point is defined as the deviation from a 7-year mean (1993–1999). AVISO has

been distributing Topex/Poseidon and ERS altimetric data worldwide since 1992, and Jason-1 and Envisat since 2002. The data has a 1/3-degree spacing with weekly temporal resolution.

The response over the ocean surface can be inferred to Ekman transport [36]. When wind stress is intense, water will move away from the typhoon center.

The horizontal transport speed can be computed from the following equations:

$$U_E = \tau_y / (\rho_{sw} f) \tag{1}$$

$$V_E = \tau_x / (\rho_{sw} f) \tag{2}$$

where $\rho_{sw}$ is the density of seawater 1027 kg·m$^{-3}$ and $f$ is Coriolis force.

In Equations (1) and (2), $U_E$ and $V_E$ are the components of the Ekman transport in the east and north directions (the unit is m$^2$·s$^{-1}$). $\tau_x$, and $\tau_y$ are the east and north wind stresses. The variations in Ekman transport during the typhoon period given in further section. It is evident that when typhoon intensity increases, wind stress increases, leading to higher Ekman transport over the ocean surface.

## 3. Results and Discussion

### 3.1. Variation in SST (Contour) and Wind (Vector) during Typhoon Haikui

During the Typhoon Haikui period, the change in SST (shaded) and wind (vectors) is illustrated in Figure 2, which includes the track with line. On 2 August, a cyclonic wind field generated obviously over the northwestern Pacific. Meanwhile, one cyclonic wind field had already landed in Taiwan, named Typhoon Saola. At the same time, Typhoon Damrey (2012) landfall occurred on the east coast of China on 2 August. Cooler SSTs persisted on the east coast of China and between Taiwan and the Chinese coast. The open ocean attained warmer SST, where Typhoon Haikui formed and cyclonic winds prevail. On 3 August, the SST around the typhoon center reached 30 °C, higher than the previous day. Haikui was upgraded to a tropical storm with a pressure of 992 hPa, and the translation speed increased on 4 August. The SST dropped by 2 °C around the typhoon center, the area of high temperature was reduced compared to the previous day, and the cyclonic winds intensified. Translation speed impacts the typhoon intensity and the SST [37]. On 5 August, the pressure decreased to 980 hPa, after 6 h, pressure intensified rapidly to 970 hPa and the translation speed reduced. Owing to the intensification, strong winds prevailed. Haikui was upgraded to a severe tropical storm when it entered the East China Sea. Small-scale and isolated spots of SST changes of 2 °C were observed to the left of the Haikui track. On 6 August, Haikui was upgraded to a typhoon with a pressure of 970 hPa. Over the study area surface water of 27.5 °C continued to expand.

From 4 to 6 August, Haikui translation speed reduced and later enhanced. There was a noticeable decrease in SST to the left of the typhoon track on 7 August, which may relate to rainfall. However, the typhoon further intensified due to passing over a warm eddy. The area, before Haikui passing showing 1 °C higher SST, cooling of SST (2 °C) occurred after Haikui passed. Usually, Typhoons produce SST cooling of more than 2 °C [38,39]. This study clearly shows SST cooling of around 3° happened, indicating that SST cooling depends on typhoon intensity. SST also has an impact on typhoon intensity and translation speed. A lower translation speed and increased SST cooling. However, the influence was not considerable, possibly due to minor variations in SST over the regions through which the Typhoon passed. Different SSTs near the typhoon center, resulting in differences in the atmospheric wind field, can cause rapid alteration in the wind field [40].

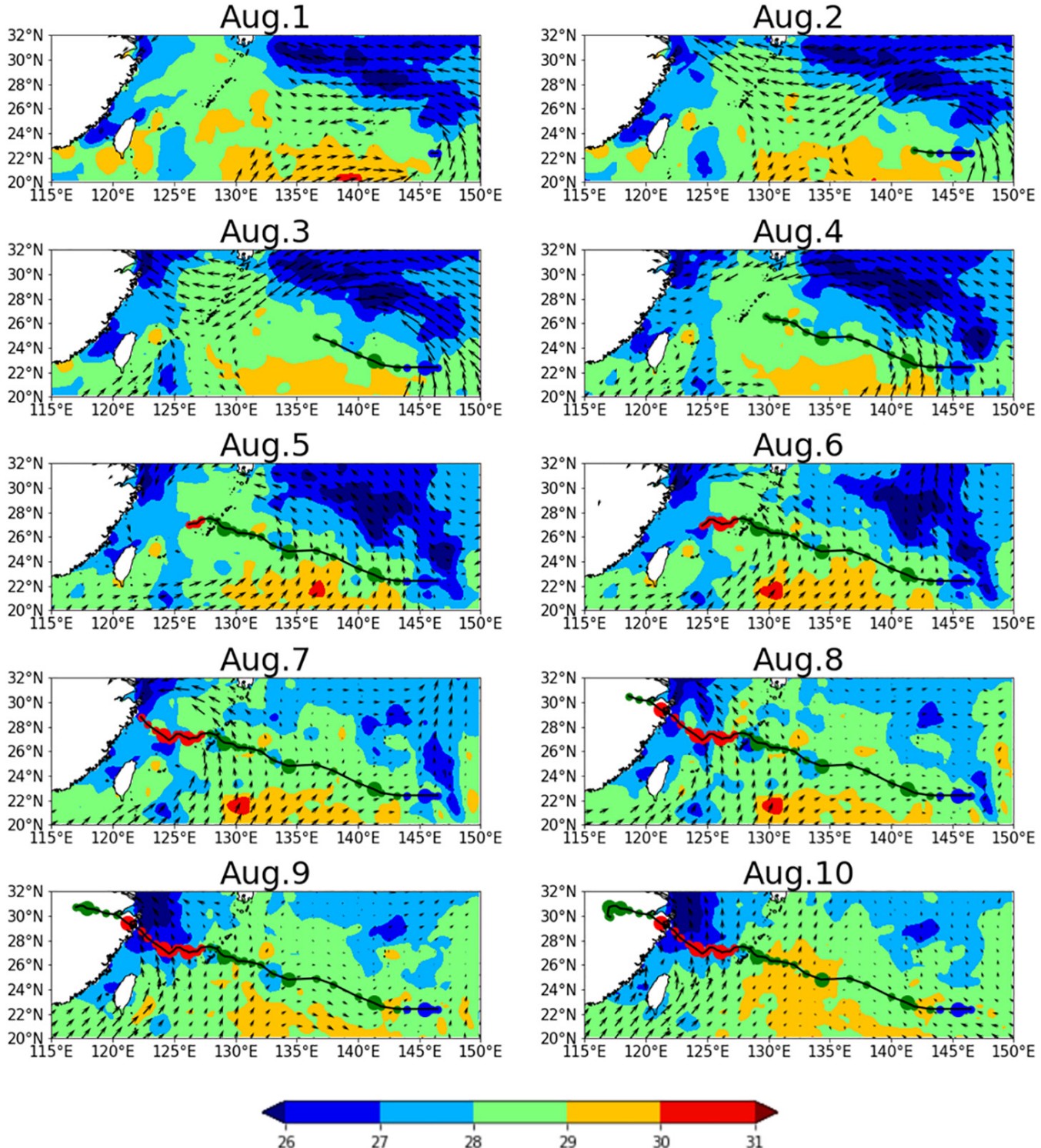

**Figure 2.** Variations in wind (vectors, m/s) and SST (shading, °C) during the study period of Typhoon Haikui. Typhoon track includes its intensity with typhoon location.

A convergence of wind fields is attributed to SST: warm sea surface can stimulate faster and stronger wind at the sea surface. With a pressure of 965 hPa, Haikui was upgraded to a severe typhoon on 7 August. In the early hours of 8 August, the Typhoon landfall occurred at Xiangshan county, Zhejiang province, China. SST cooling (2.5 °C) was observed in the coastal area on the right side of the typhoon track. On 9 August, the water along the coast decreased by more than 3 °C with an extension of the region. On 10 August,

the typhoon-affected area was warming and increased to 28.5 °C SST. The SST increased continuously on 11 August and reached 29 °C in the open ocean. Throughout the study period, SST cooling was observed on the right side of the track with maximum cooling of 3 °C, mainly because of strong atmospheric wind prevailing over the ocean surface to produce strong upwelling. The cool temperatures persisted for more than two days, then the SST increased slowly. The atmosphere–ocean interaction process leads to a gradual increase in the SST.

*3.2. Dynamic Response of Atmosphere–Ocean Parameters and Relations between them*

The dynamic relations between atmospheric parameters, such as wind and rain, and ocean parameters, such as SST, Ekman transport, and SSH, were studied and presented. During typhoon periods, cyclonic winds prevail, and the wind becomes higher as the intensity of the typhoon increases. The horizontal force of the wind on the sea surface is called wind stress [41]. Figure 3 depicts the wind stress variations during Typhoon Haikui over the study area, and the effect of wind on the ocean surface (Ekman transport) is shown in Figure 4. The overlying wind stress induces changes in SST and atmospheric stability depending on the SST gradient and the wind direction relative to the SST gradient vector [42]. Cyclonic winds produce wind stress curl to the right of the storm. On 2 August, there were two main wind stress centers. One, around Taiwan, was caused by Typhoon Saola, while the other, far away from China's coast, was due to Typhoon Haikui. From 3 to 5 August, the wind stress of Typhoon Haikui decreased as the center moved towards the western approach to eastern China. From 6 August to 7 August, the typhoon's intensity increased. The wind stress curl generates upwelling over the open ocean through Ekman pumping [43,44]. Curl-driven Ekman upwelling in mesoscale model wind fields is comparable to coastal upwelling [45]. However, on and after 8 August, wind stress cannot be seen over the ocean and in coastal waters. We have chosen wind stress data over the open ocean, so the land data is missing. According to the record of Typhoon Haikui, after it landed in Xiangshan county on early 8 August, its intensity declined.

On 2 August, we observed two intense areas of Ekman transport, one at the Typhoon Haikui generated area and another on the Taiwan coast where Typhoon Damery made landfall. However, the intensity of Ekman transport was not predominant. When we compare Figure 4 with Figure 2, lower SSTs were found in the areas of Ekman transport. On 3 August, the Ekman transport was due to Typhoon Haikui. On 4 August, the Ekman transport enhanced because the typhoon intensity increased, SST cooling happened, and the extent of the area extent increased. On 5 August, it was the first time Ekman transport had appeared on both sides of the typhoon track due to the increase in Haikui's intensity. In the northern hemisphere, the water mainly moves to the right side of the wind due to the Coriolis effect. However, there was more transport to the right side of the typhoon track than on the left side, which indicates that upwelling and SST cooling due to the typhoon's passage above the ocean. On 6 August, the intensity of Ekman transport increased, leading to a strong upwelling. Meanwhile, the volume of transport was less than the day before. On 7 August, Typhoon Haikui entered the coastal waters of eastern China.

Owing to the higher intensity of the typhoon, a strong upwelling and cooler SSTs occurred in the coastal waters (Figure 1). However, there was an obvious drop in SST on the left of the typhoon track. After the Typhoon's landfall on 8 August, there was SST cooling along the coastal area of eastern China. Ekman transport brought the subsurface colder water to form an upwelling. A noticeable change in SST could be observed for several days after typhoon landfall because the rate of upwelling was slow ($10^{-5}$ m/s). On 9 August and 10 August, SST near the coastal area dropped more than 3 °C while the volume transport was not so apparent. This was a delayed effect caused by slower upwelling. On 11 August, SST over the ocean areas increased to 29 °C, and there was no specific Ekman transport observed over the ocean.

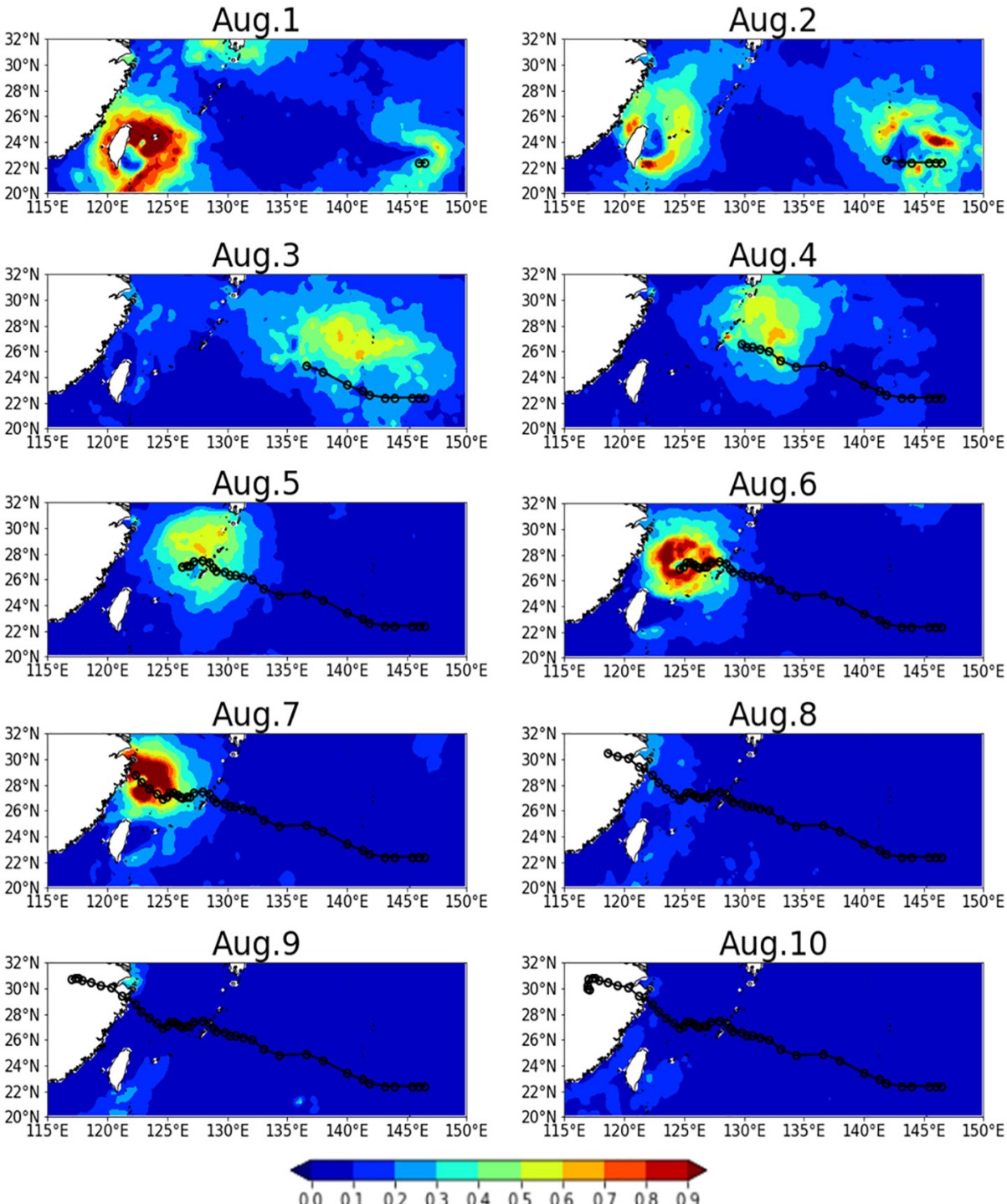

**Figure 3.** Wind stress variations (Pascal) during Typhoon Haikui, including the typhoon track (every 6 h). Wind stress intensity is shown with a color code (dark blue represents no wind stress, red represents higher intensity).

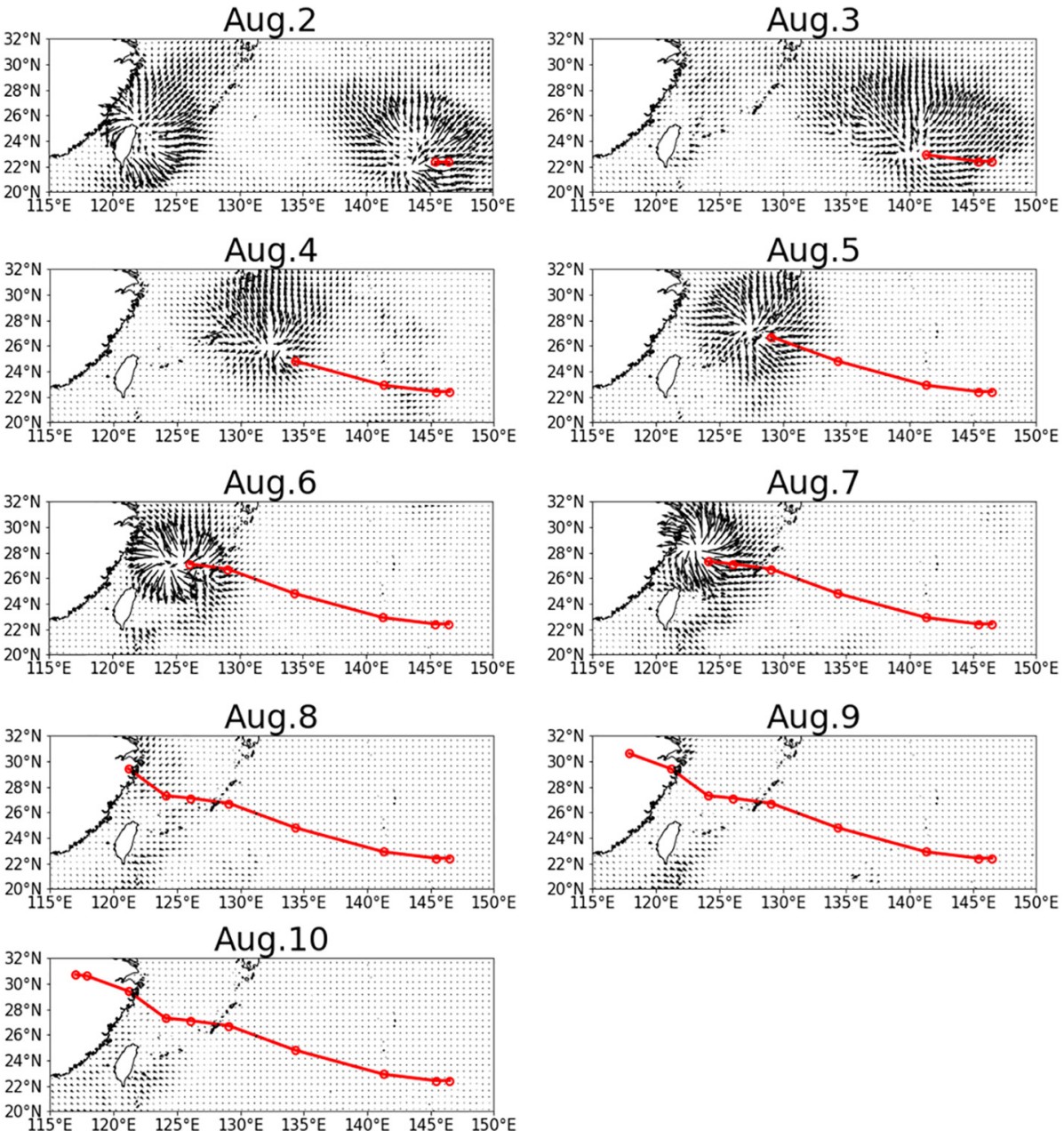

**Figure 4.** Ekman transport variations (m/s) during Typhoon Haikui. Typhoon track included as red line, circles represents the location on that day.

Figures 2–4 indicate the relationship between the SST, wind, wind stress, and Ekman transport. Wind stress curl induces Ekman pumping in the open ocean, and typhoon-induced inertial oscillations accompany vertical mixing or entrainment [46,47]. The Ekman pumping induced by typhoons causes the decline in SST [48]. When the typhoon intensity increases (pressure decreases and wind increases), wind stress increases to push the surface waters away from the center, which is evident from the Ekman transport. Intense wind stress and Ekman transport induce strong upwelling leading to SST cooling over the ocean and in coastal waters. The distinct feature of the upper ocean response to a moving typhoon is SST cooling. Typhoon formation due to higher wind stress causes intense mixing and divergence between the surface and subsurface of the ocean. Through entrainment deepening the mixed layer, induced upwelling forces surface water redistribution, and the cold bottom water comes to the surface, so SST cooling happens.

*3.3. Cumulative Precipitation of Haikui Typhoon Period*

The calculated cumulative rainfall is presented in Figure 5 using GPCP merged precipitation data during the Typhoon Haikui period (Daily variation of rainfall (Figure S1) given in Supplementary material). Generally, the heaviest precipitation occurs in front of a typhoon [49]. The asymmetry in rainfall varies with typhoon intensity [50]. From 2 to 11 August, three typhoons affected the study area. Two typhoons (Saola and Damary) made landfall on China's coast and, at the same time, Typhoon Haikui was in the ocean. Typhoon Haikui produced rainfall of 275 mm over Zhejiang province during the typhoon period, which reveals that precipitation increases after typhoon landfall. There are two reasons for this: (1) after landing, the typhoon changed its direction towards the west and suddenly paused; (2) the typhoons had a long duration, and intensity declined slowly with a slow translation speed over the land [51]. From Figure 5, the rainfall on the left side of the track was higher than that on the right. Typhoon Haikui generated heavy precipitation, especially over the Huangshan mountains, which received a record rainfall of 678 mm. Tahereh et al. [12] found that precipitation rate increases after a typhoon on a large scale. Over the study area, there was rainfall on both sides of the typhoon track, but the left side of the typhoon track received more than the right, which is similar to Typhoon Kaemi [15].

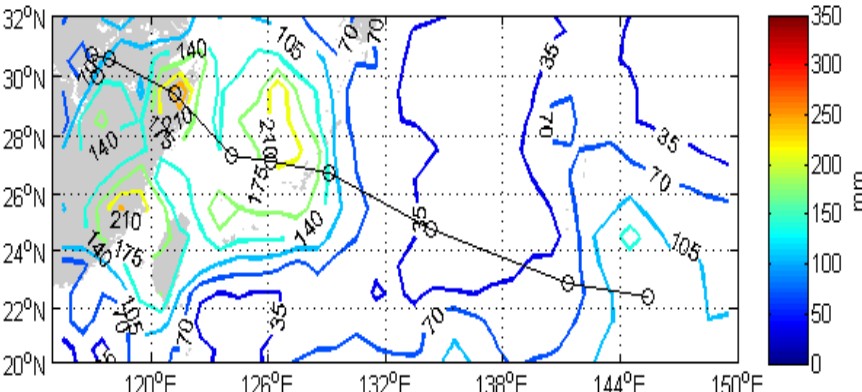

**Figure 5.** Cumulative rainfall (GPCP) over the study area during Typhoon Haikui. Typhoon track included in black line. Color contours represent cumulative rainfall.

Typhoon translation speed can have a significant effect on the asymmetric distribution of typhoon rainfall [52]. Heavier precipitation typically occurred away from the center, thus bearing the common feature of a weak typhoon. Translation speed reduced during Typhoon Haikui's landfall, and higher rainfall occurred. Furthermore, three main precipitation areas were near the mainland. The topography of land and convergence of typhoon peripheral wind also results in robust air–sea interaction induced seawater evaporation. SST variations in the typhoon area are less and can lead to a significant change in precipitation. However, different factors also affect precipitation variations, and SST is one of them. SST increases before a typhoon, whereas the precipitation rate increases after a typhoon's landfall, which means that the increased SST before typhoon occurrence prepares the required energy and brings additional precipitation during the typhoon [11]. A warm (or cold) sea surface can stimulate faster and strong (or slower and weaker) wind at the sea surface [3,53].

*3.4. Post Dynamic Effect of Typhoon Haikui on the Ocean Surface*

The post dynamics of Typhoon Haikui affected in SSH (from AVISO) and chlorophyll-a concentration (MODIS). The SSH and chlorophyll data are weekly and depict the SSH variation before, during, and after the Haikui Typhoon period in Figures 6 and 7. Figure 6a illustrates the SSH before Typhoon Haikui. Figure 6b represents the SSH during the Typhoon period, and Figure 6c shows the SSH after the typhoon. On 1 August, a tropical depression (Haikui) had just formed over the ocean, higher SSH observe over the coastal waters of China due to Typhoon Saomi. On 8 August, Typhoon Haikui landed on the

east coast of China. On the right side of the Typhoon track, SSH at the coast is 33 cm. However, on the left side of the Typhoon track SSH is 11 cm. One can observe that the SSH is higher in the coastal areas of Figure 6a,c, depicting the typhoon landfall on China's coast, however SSH decreased in the open ocean area. With the daily data, daily variations can be observed.

The force of the typhoon wind stress (please refer to Figure 3 on 7 August) and Ekman transport (please refer to Figure 4, on 7 and 8 august) happening towards the right side of the typhoon track gives a clear picture of why SSH was higher in the coastal region and on the right side of the typhoon track. Surface water flows away from the typhoon center through divergence, which then lowers the SSH below the typhoon center [54–56]. Away from the storm center, the wind stress drives convergent Ekman transport and positive SSH [25]. The anomalies are higher on the right side of the typhoon track due to the higher wind stress curl. During the passage of the typhoon, SSH is notably decreased.

Furthermore, to analyze the post- typhoon effects on chlorophyll concentration, MODIS Aqua-Weekly Ocean Color data on chlorophyll-a concentration were obtained; the chlorophyll data are plotted and presented in Figure 7. Figure 7a describes the chlorophyll variations before the typhoon (27 July), Figure 7b during the typhoon period (4 August), and Figure 7c illustrates the post-typhoon (12 August) period. Chlorophyll concentrations were high along the coastal area in Figure 7a,c, representing before and after the typhoon's landfall. Before the Haikui Typhoon landed, the chlorophyll concentration was high along the shore due to the landfall of another typhoon, Damrey, and then decreased during the Typhoon Haikui period; later chlorophyll became high again after the Haikui landfall on 12 August. Typhoons have a delayed effect on chlorophyll concentrations, with an average delay of 5.94 days [19].

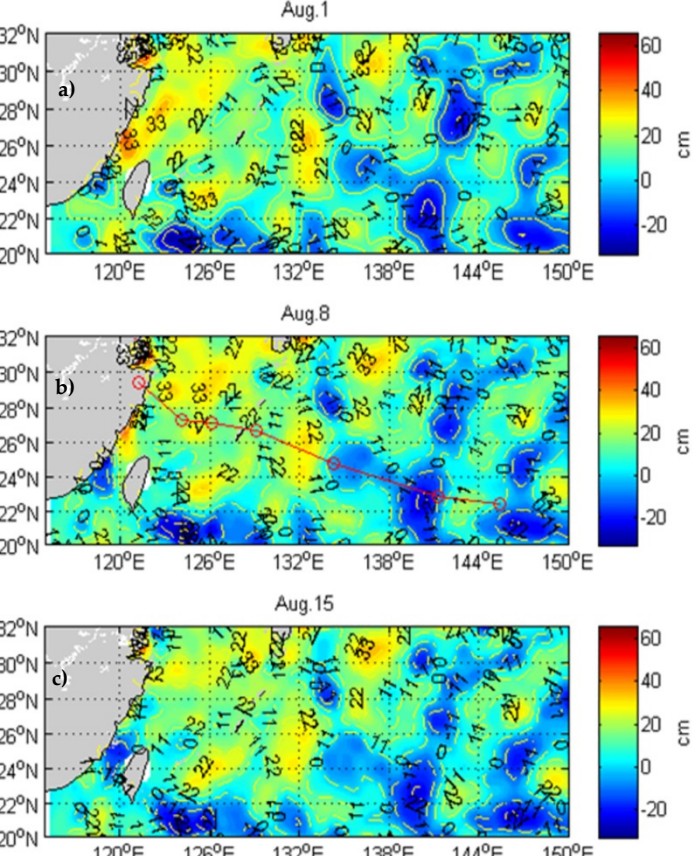

**Figure 6.** SSH (Sea surface height) variations during Typhoon Haikui. (**a**) Before the typhoon (1 August), (**b**) during the typhoon (8 August), and (**c**) after the typhoon (15 August).

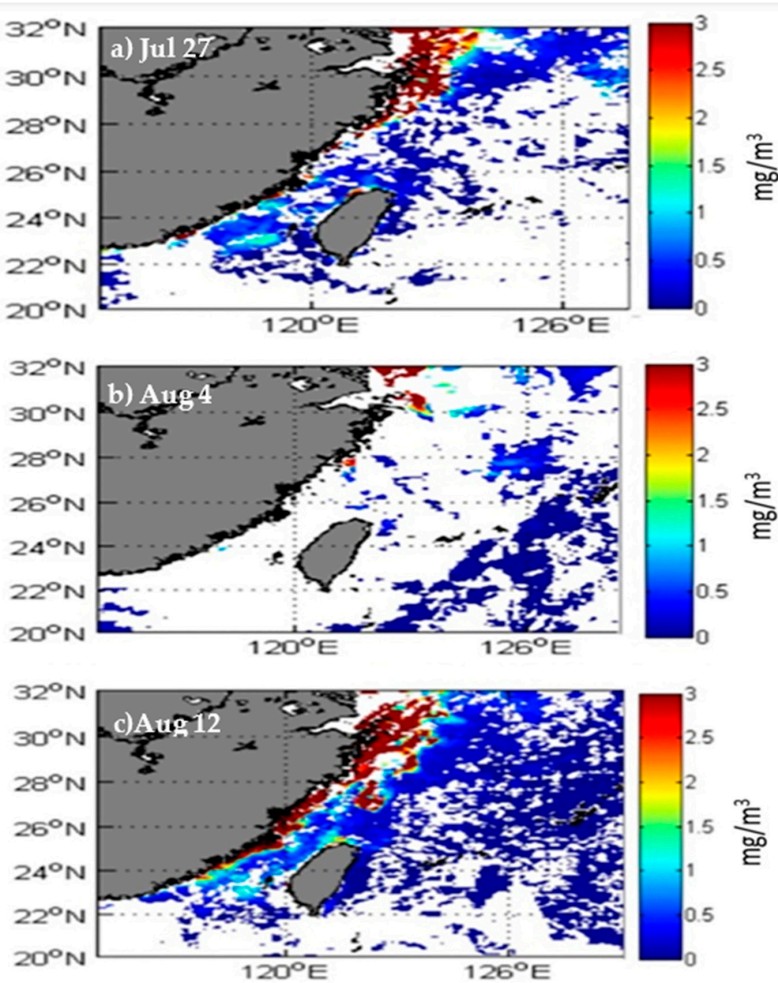

**Figure 7.** Variation in chlorophyll-a concentrations near coastal areas during Typhoon Haikui. (**a**) Before (July 27), (**b**) during (August 4) and (**c**) after the typhoon (August 12).

Phytoplankton relies on chloroplasts to conduct photosynthesis, which occurs in the euphotic zone. In summer, the sea area (25.2° N, 120° E, and 27.2° N, 120.5° E) off the coast of Fujian and Zhejiang has strong upwelling. In general, the rising of subsurface water is too small to provide rich nutrients for phytoplankton. Thus, the sea surface euphotic layer has had a poor nutritional condition for a long time. During the typhoon, the large eddy and low pressure are caused by itself, resulting in the sea surface pressure of the typhoon zone being lower than that of the surroundings. Accordingly, this region forms a strong upwelling and strengthens the vertical mixing of seawater. This effect transports phosphate and nitrate from the deep cold-water region to the euphotic zone. So, the phytoplankton will get enough nutrients to grow dynamically with a corresponding increase in chlorophyll concentration. However, often in rainy weather during a typhoon, vertical mixing and breaking of the seawater reduce the transparency of the seawater. Phytoplankton cannot grow well because of a lack of sunlight, which limits the increase in chlorophyll concentration to a certain extent [26]. In contrast, after the passage of a typhoon, the vertical mixing of seawater and the upwelling are stable. At a specific time, the concentrations of the nutrients on the sea surface appeared to be relatively stable. However, the typhoon could not promote the unlimited growth of chlorophyll.

Chlorophyll concentration will increase after the passage of a typhoon. The bottom figure has a higher chlorophyll concentration than the top one. After the passing of Haikui, the MODIS image composite illustrates an enhancement of phytoplankton bloom. of the increase in chlorophyll is due to the upwelling and mixing, which brings subsurface nutrient-rich water to the sea surface. As mentioned in the previous sections, due to robust

wind stress over the coastal area, Ekman transport increases, causing intense upwelling near the coast, and SST cooling (due to upwelling) also happens. SST along the coastal area was lower during the typhoon passage over the study period. After Typhoon Haikui landfall occurred, rainfall increased and persisted for a few more days to produce higher river discharge. River discharge brings a lot of sediment to the coastal waters and can be a source of nutrients [57], and when sediment is disturbed in the coastal area, nutrients are released [58]. However, after Typhoon Haikui, heavy rainfall occurred, and terrestrial runoff to the coastal waters enhanced the chlorophyll. The chlorophyll concentration along the coastal area was higher. However, low temperature during the typhoon (due to entrainment) leads to booms in primary productivity, which contribute to the alongshore phytoplankton blooms.

Variations in SST also play a significant role in chlorophyll concentration by improving marine primary productivity and adjusting the marine ecosystem, contributing a lot to the alongshore phytoplankton blooms, fishery increases, carbon cycling, etc. Better technology and higher resolution satellite data could improve the accuracy of measuring parameters such as SSH and chlorophyll.

## 4. Conclusions

Remote sensing and reanalysis data were used to study the dynamic variations during Typhoon Haikui. Strong wind caused by typhoons leads to the horizontal transportation of seawater. Ekman transport leads to the upwelling and cooling of SST. The dynamic processes occurring during the Typhoon Haikui period and the post-typhoon period are concluded as follows:

From Figures 2–4, the cyclonic wind increased to typhoon intensity. The wind stress exerted over the sea surface produces Ekman transport leading to coastal upwelling to cause in cooling of SST in coastal areas. Maximum SST cooling of 3 °C occurred on the right side of Typhoon Haikui. Haikui attained its highest intensity when the translation speed was low, producing more wind stress with higher Ekman transport. After landfall, Typhoon Haikui's moved slowly over the land, producing heavy rainfall of 275 mm over Zhejiang province. After Haikui's landfall, due to upwelling and river discharge due to heavy precipitation over the coastal area, the chlorophyll blooms increased. An increase in SSH can be seen post-landfall where the Ekman transport occurred on 7 and 8 August. Overall, this study indicates a dynamic relationship between the atmosphere and ocean parameters during the typhoon. This work will continue to analyze air–sea interaction parameters.

**Supplementary Materials:** The following are available online at https://www.mdpi.com/article/10.3390/atmos14030518/s1, Figure S1: GPCP rainfall variations during Typhoon Haikui study period. The typhoon track included different days represented as circles. Colour contours represent the rainfall and the amount of rainfall given in the contours. Also, references [15,49] used in Supplementary Material file.

**Author Contributions:** M.V.S., B.G. and L.W. conceptualized the study; M.V.S. and W.Z. implemented work and associated software; M.V.S., B.G. and W.Z. made formal analyses; L.W. and B.G. provided computational resources; L.W. acquired funding; all authors participated in review and edited the text. All authors have read and agreed to the published version of the manuscript.

**Funding:** This work is funded by Tsinghua university (sklhse-2021-B-01) and Chinese Academy of Sciences (2020-ZW11-A-023).

**Data Availability Statement:** Publicly available datasets were analyzed in this study.

**Acknowledgments:** Authors express their thanks to NOAA for SST Products, AVISO for SSHA, MERRA for Wind data, ASCAT for Wind stress data, MODIS Aqua for Weekly chlorophyll data and GPCP for daily precipitation data, which is freely available for research.

**Conflicts of Interest:** The authors declare no conflict of interest.

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
