# Peer review of "Dynamic Response of Atmospheric and Ocean Parameters and Their Relation to Typhoon Haikui (2012) Using Satellite Data"

_atmosphere, doi:10.3390/atmos14030518_

Round 1
Reviewer 1 Report
The authors have performed a case study of Typhoon Haikui (2012) in which they examined how various atmospheric and oceanic parameters evolved over time due to the passage of the typhoon. I have attached my full annotated statements, but here are my general statements:
1. There are many grammatical errors associated with this manuscript (such as sentence fragments, subject-verb agreement, and awkward phrasing). There must be a substantial improvement in the grammatical style before this paper is ready for publication.
2. It appears that the authors have used MATLAB to prepare the figures for this paper. However, the image resolution of these figures are too low for publication. I suggest the figures are re-produced for higher quality.
3. The biggest weakness of this paper involves the explanations and the conclusions. Much of this paper deals with Ekman transport phenomena and the impact of storm translation; however, there are very few papers that are referenced that examines boundary layer dynamics of tropical cyclones. The authors do not address the asymmetries associated with storm motion, along with the nonlinear advective interactions that lead to asymmetric boundary layer flow. Furthermore, the discussion on why there is an asymmetric pattern of rainfall should be discussed in much more detail. I would advise the authors to supplement their observational work with numerical model simulations that demonstrate the relationships between Ekman transport and SST.

Author Response
We thank the reviewer for his comments.
We modified the manuscript as per the reviewer's comments.
Details are given in the file.

Reviewer 2 Report
This paper reports atmospheric and oceanic variables from remote-sensing and reanalysis data, corresponding to the typhoon Haikui.
The analysis is rather straightforward, and main results are a sequence of figures illustrating the wind field, wind stress, Ekman transport, SST, chlorophyll concentration, ... The amount of analysis performed is rather limited, but the results and discussion are interesting.
The conclusions of the study are reasonable, but perhaps not surprising.
The manuscript needs to be read and corrected by a native English speaker, for clarity and conciseness of description.
Author Response
We thank the reviewer for his valuable comments.
We modified the manuscript as per the comments.
Details are given in the file.

Author Response

(The authors gave the same response as above.)

Reviewer 4 Report
The paper presented studies the air-sea interactions at the development of Typhoon Haikui. The work is interesting, but it is very poorly written, confused, the figures are of very low quality and the bibliography has no international character. There are many changes (important, and structural) to be made, and I do not consider the paper ready for publication. I suggest major revision.
Here are the major suggestions
1) The quality of the figures is low. The figures are poorly laid out, vertically elongated. The color scales are "saturated" and this does not allow to understand the rapid variations of the parameters studied. I suggest, in general, to use scales ranging from "white" to colored, for the variables between very low values, and its maximums. Use color scales with white "zero" for differences.
2) In the figures, reduce the size of the vectors, increase their quantity.
3) In figure 1 show the intensity of the cyclone, not just the trajectory.
4) Figure 4, using a different color scale (more suitable and with the white zero, (or the minimum white value), use the contour + the shaded. In this way the plot will be more readable. Add, to the contour, the label of the values.
5) Figure 5, as in figure 4. Figure 6, is incomprehensible. Use another color scale (with white zero), add the contour, reduce the font of the labels.
6) Figure 7, incomprehensible, is saturated, only the minimum and maximum values ​​are seen.
7) The bibliography has no international relevance. I suggest some works that can improve bibliolography, which is too short and limited to national authors, although these themes have been treated by many authors.
8) Add citations to all datasets used.
9) The introduction is not representative of the study, it does not clarify the state of the art on this issue, it is written in a very summary way. Extensive review and writing of the introduction and on the purposes of the work is required.
10) The descriptions of the images are very short. They must be expanded, a lot, and must describe in detail what the image shows.
11) Add units of measurement in the images (on the side of the color bar)
12) The sentence of line 36, how is it contextualized in the article and in this application? To strengthen this theme I suggest adding papers that confirm your theory:
Olabarrieta, M.; Medina, R.; Castanedo, S. Effects of wave-current interaction on the current profile. Coast. Eng. 2010, 57, 643–655
Warner, J.C.; Sherwood, C.R.; Signell, R.P.; Harris, C.K.; Arango, H.G. Development of a three-dimensional, regional, coupled wave, current, and sediment-transport model. Comput. Geosci. 2008, 34, 1284–1306.
Zambon, J.B.; He, R.; Warner, J.C. Investigation of hurricane Ivan using the coupled ocean-atmosphere-wave- sediment transport (COAWST) model. Ocean Dyn. 2014, 64, 1535–1554.
Ricchi, A., Bonaldo, D., Cioni, G. et al. Simulation of a flash-flood event over the Adriatic Sea with a high-resolution atmosphere–ocean–wave coupled system. Sci Rep 11, 9388 (2021). https://doi.org/10.1038/s41598-021-88476-1
Warner, J. C., Armstrong, B., He, R. & Zambon, J. B. Development of a coupled ocean–atmosphere–wave–sediment transport (COAWST) modeling system. Ocean Model 35(3), 230–244. https://doi.org/10.1016/J.OCEMOD.2010.07.010 (2010).
Meroni, A. N., Parodi, A. & Pasquero, C. Role of SST patterns on surface wind modulation of a heavy midlatitude precipitation event. J. Geophys. Res. Atmos. 123, 9081–9096. https://doi.org/10.1029/2018JD028276 (2018).
and others...
13) The article, which talks about air-sea interaction, does not mention fundamental articles that describe the complex air-sea interactions. This part is poorly described in the introduction, and it is very badly done. I suggest rewriting and adding this basic references:
Charnock, H. Wind stress on a water surface. Q. J. R. Meteorol. Soc. 1955, 81, 639–640.
Meroni, A. N., Parodi, A. & Pasquero, C. Role of SST patterns on surface wind modulation of a heavy midlatitude precipitation event. J. Geophys. Res. Atmos. 123, 9081–9096. https://doi.org/10.1029/2018JD028276 (2018).
O’Neill, L. W. et al. Observations of SST-induced perturbations of the wind stress field over the Southern Ocean on seasonal timescales. J. Clim. 16(14), 2340–2354. https://doi.org/10.1175/2780.1 (2003).
O’Neill, L. W. et al. High-resolution satellite measurements of the atmospheric boundary layer response to SST variations along the Agulhas return current. J. Clim. 18(14), 2706–2723. https://doi.org/10.1175/JCLI3415.1 (2005).
O’Neill, L. W. et al. The effects of SST-induced surface wind speed and direction gradients on midlatitude surface vorticity and divergence. J. Clim. 23(2), 255–281. https://doi.org/10.1175/2009JCLI2613.1 (2010).
Chelton, D. B. & Chelton, D. B. The impact of SST specification on ECMWF surface wind stress fields in the Eastern Tropical Pacific. J. Clim. 18(4), 530–550. https://doi.org/10.1175/JCLI-3275.1 (2005).
Chelton, D. B. et al. Observations of coupling between surface wind stress and sea surface temperature in the Eastern Tropical Pacific. J. Clim. 14(7), 1479–1498. https://doi.org/10.1175/1520-0442(2001)014%3c1479:OOCBSW%3e2.0.CO;2 (2001).
Chelton, D. B. et al. Summertime coupling between sea surface temperature and wind stress in the California current system. J. Phys. Oceanogr. 37(3), 495–517. https://doi.org/10.1175/JPO3025.1 (2007).
Obermann,A.,Edelmann,B.&Ahrens,B.InfluenceofseasurfaceroughnesslengthparameterizationonMistralandTramontane simulations. Adv. Sci. Res. 13, 107–112. https://doi.org/10.5194/asr-13-107-2016 (2016).
Author Response
We thank the reviewer for his valuable comments
We modified the manuscript as per the comments.
Details are given in the file.

Round 2
Reviewer 1 Report
I would like to thank the authors for substantially improving their manuscript. The authors have addressed many of my original concerns. There are some grammatical concerns that should be addressed, which I've attached as a PDF. I think that the manuscript is suitable for publication.

Author Response
We thank you for your appreciation, we did our level best to modify the paper.
We have modified the manuscript as you mentioned.

Reviewer 3 Report
Please see the attached file.

Author Response
We thank you for your comments, we revised the paper as per your comments.
We modified the numbers in the methodology and results.

Reviewer 4 Report
As per the previous review, I confirm what was proposed for this work.
The critical issues highlighted in the first review have remained the same and impact on 3 fundamental characteristics for the publication of a scientific paper.
- the quality of the figures is very low. They are vertically deformed. The color scales are saturated, they do not facilitate the identification of the structures. the lables cover the lines and the vectors cover the labels. The figures are unpresentable.
- the abstract is very short, does not clarify and does not adequately describe the work. The introduction confusingly discusses the state of the art. I had asked to add a part discussing air-sea interactions, it was not done. I had also asked to expand the quotes, it was done but randomly.
- The "materials and methods" section must also implement the numerical techniques used. The discussion of these techniques is brief and poorly supported by bibliography.
Summing up :
1) absolutely improve the quality of the figures.
2) expand the discussion of air-sea interactions, add all the quotes I suggested, and more.
3) adequately describe the numerical methods applied in this work.
Author Response
We appreciate your comments, however, we already modified the figures with higher quality.
Answers to your comments are given in the word file attached.
